

# Differential Absorption Radar Techniques: Water Vapor Retrievals

Luis Millán[1], Matthew Lebsock[1], Nathaniel Livesey[1], and Simone Tanelli[1]

[1]Jet Propulsion Laboratory, California Institute of Technology, Pasadena, California, USA.

*Correspondence to:* L. Millán (lmillan@jpl.nasa.gov)

**Abstract.** Two radar pulses sent at different frequencies near the 183 GHz water vapor line can be used to determine total column water vapor and water vapor profiles (within clouds or precipitation) exploiting the differential absorption on and off the line. We asses these water vapor measurements by applying a radar instrument simulator to CloudSat pixels and then, running end-to-end retrieval simulations. These end-to-end retrievals enable us to fully characterize not only the expected precision but

also their potential biases, allowing us to select radar tones that maximize the water vapor signal minimizing potential errors due to spectral variations in the target extinction properties. A hypothetical CloudSat-like instrument with 500 m by ∼1 km vertical and horizontal resolution and a minimum detectable signal and radar precision of -30dBZ and 0.16dBZ, respectively, can estimate total column water vapor with an expected precision of around 0.03 cm with potential biases most of the time smaller than 0.26 cm, even under rainy conditions. The expected precision for water vapor profiles was found to be on average

around 89% with potential biases most of the time smaller than 77% when the profile is being retrieved close to surface but smaller than 38% above 3 km. By using either horizontal or vertical averaging, the precision will improve vastly with the measurements still retaining a considerably high vertical and/or horizontal resolution.

©2016 California Institute of Technology. Government sponsorship acknowledged.

## 1 Introduction

The WMO (2014) statement of guidance for global numerical weather prediction concluded that one of the critical atmospheric variables that are not adequately measured by current or planned systems is humidity, in particular, profiles with adequate vertical resolution in cloudy areas were recommended (Anderson, 2014). Due to its importance, several space-borne methods have been used to observe atmospheric water vapor, such as, passive near-infrared or microwave imaging, passive infrared or microwave sounding, and radio occultation techniques. Most of these techniques have been shown to improve weather

forecasting performance once assimilated (Anderson, 2007) and are operationally used, but each has limitations. For example, infrared and microwave sounders have broad weighting functions near the Earth's surface, which considerably limit their vertical resolution. Near-infrared and microwave imaging can only provide column water vapor and hence, they do not provide any information on its vertical distribution. Additionally, near-infrared and infrared techniques cannot penetrate cloudy scenes. Lastly, even though radio-occultation techniques are sensitive even to the boundary layer water vapor burden, atmospheric

ducting effects associated with the top of the boundary layer limit their accuracy (Ao et al., 2003).





In this study we asses the differential absorption radar (DAR) concept to profile water vapor in cloudy and rainy areas. This concept exploits the difference between the radar reflectivity at different frequencies ("on" and "off" an absorption line) to estimate the absorbing gas path between the radar and the scattering target. The DAR technique can be used to retrieved surface pressure by estimating the column of oxygen using frequencies near the 60 GHz absorption band (Flower and Peckham, 1978; Lin and Hu, 2005; Lawrence et al., 2011; Millán et al., 2014). Prior studies show that this technique can be used to retrieve water vapor profiles using three frequencies centered around the 22 GHz water vapor absorption line (Meneghini et al., 2005), frequencies at 10 and 94 GHz (Tian et al., 2007) and at 2.8 and 35 GHz (Ellis and Vivekanandan, 2010) using the water vapor continuum. In addition, Lebsock et al. (2015) assessed the feasibility of water vapor sounding using two frequencies at the wings of the 183 GHz water vapor line using large eddy simulations (LES). These high resolution LES allowed the assessment of uncertainties due to small scale heterogeneities within the radar field of view (commonly known as non-uniform beam filling) as well as uncertainties due to the particle size distribution. However they do not provide context on the capabilities of DAR over a wide variety of Earth's clouds. This global climatological context is best provided by observations. Following Lebsock et al. (2015) the study will focus on the wing of the 183 GHz water vapor line, but we will evaluate the DAR capabilities using global cloud observations from CloudSat (Stephens et al., 2002). We specifically focus on a spaceborne observational platform.

A spaceborne implementation of a water vapor DAR would provide observational capabilities that complement existing remote sensing methods. In particular the method specifically samples within the cloudy environment. this could provide much needed observations within the poorly sampled cloudy boundary layer and help constrain the relative humidity in ice clouds. Furthermore, the method can provide high spatial resolution column water vapor in all weather conditions and over all surfaces.

In this study we evaluate the ability of the DAR technique to retrieve water vapor with high vertical and horizontal resolutions under cloudy/rainy conditions and we diagnose which radar tones are better suited for sampling different altitude ranges. This paper is organized as follows: the measurement theory is described in section 2, the radiometric model is described in section 3, total column water vapor retrievals are discussed in section 4 while section 5 explores the profiling capabilities of this technique. Section 6 summarize the results.

## 2 Theoretical basis

As shown by Lebsock et al. (2015), the ratio of two radar reflectivities, neglecting multiple scattering, can be expressed as

$$\frac{Z(\nu_1,r)}{Z(\nu_2,r)} = \frac{\Upsilon^2(\nu_1,r)\,\eta(\nu_1,r)\,\lambda_1^4}{\Upsilon^2(\nu_2,r)\,\eta(\nu_2,r)\,\lambda_2^4}\,\frac{|K(\nu_2,r)|^2}{|K(\nu_1,r)|^2} \tag{1}$$

where $\nu$ is a radar tone frequency, $\lambda$ is the wavelength of radiation, and $K(\nu,r)$ is the dielectric constant of the target, $\eta(\nu,r)$ represents the hydrometeors backscatter coefficients, and $\Upsilon^2(\nu,r)$ is the two-way transmission along the range $r$ given by,

$$\Upsilon^2(\nu,r) = \exp\left(-2\int_0^r [\sigma_{\text{gas}}(\nu,r) + \sigma_{\text{Pext}}(\nu,r)]\,\mathrm{d}r\right) \tag{2}$$





where $\sigma_{\mathrm{gas}}(\nu, r)$ represents the gaseous absorption coefficient and $\sigma_{\mathrm{Pext}}(\nu, r)$ the particulate extinction (the sum of absorption and scattering) coefficient. Note that in equation 1, when the scattering target is the surface rather that hydrometeors along the path; $\eta(\nu, r)$ is replaced by the normalized surface cross section $\sigma_0(\nu)$.

Assuming that these frequencies are chosen close to a strong absorption line, the frequency dependence of $\sigma_{\mathrm{Pext}}(\nu, r)$, $\eta(\nu, r)$ and $\sigma_0(\nu)$ is small relative to that of $\sigma_{\mathrm{gas}}(\nu, r)$ (see Figure 1), equation 1 simply becomes

$$\frac{Z(\nu_1, r)}{Z(\nu_2, r)} = \frac{\Upsilon^2(\nu_1, r)}{\Upsilon^2(\nu_2, r)} = \exp\left(-2\int_0^r [\sigma_{\mathrm{gas}}(\nu_1, r) - \sigma_{\mathrm{gas}}(\nu_2, r)]\, \mathrm{d}r\right) \tag{3}$$

which can be re-written as,

$$\frac{Z(\nu_1, r)}{Z(\nu_2, r)} = \exp\left(-2\int_0^r \rho(r)\sum_i v_i(r)\left[\kappa_i(\nu_1, r) - \kappa_i(\nu_2, r)\right]\, \mathrm{d}r\right) \tag{4}$$

where $\rho(r)$ is the air density and the sum is over all the absorbers with monochromatic absorption coefficient $\kappa_i(\nu, r)$ and volume mixing ratio $v_i(r)$.

Furthermore, close to a strong absorption line, the monochromatic absorption coefficient for the rest of the absorbers (at two close enough frequencies) are similar, leaving mostly the influence of the main absorber. For example, next to the $183\,\mathrm{GHz}$ $\mathrm{H_2O}$ absorption line, equation 4 can be simplified as,

$$\frac{Z(\nu_1, r)}{Z(\nu_2, r)} = \exp\left(-2\int_0^r \rho(r)\, v_{\mathrm{H_2O}}(r)\left[\kappa_{\mathrm{H_2O}}(\nu_1, r) - \kappa_{\mathrm{H_2O}}(\nu_2, r)\right]\, \mathrm{d}r\right) \tag{5}$$

which expressed in decibels relative to $Z$ units ($\mathrm{dB}Z$) and, using the ideal gas law, results in

$$\mathrm{dB}Z(\nu_1, r) \; - \; \mathrm{dB}Z(\nu_2, r) \; \propto u_{\mathrm{H_2O}} = \int_0^r \frac{p(r)}{RT(r)}\, v_{\mathrm{H_2O}}(r) \tag{6}$$

where $R$ is the gas constant, $p$ is pressure and $T$ is temperature.

Equation 6 shows that difference between two radar tones expressed in $\mathrm{dB}Z$ units is proportional to the partial water vapor path $u_{\mathrm{H_2O}}$ between the radar and the scattering target. This means that a range-gated radar may be used to estimate profiles of water vapor density inside cloudy or rainy profiles assuming a temperature and pressure profile (e.g. from reanalysis fields). Furthermore, this technique may be used to estimate the total column water vapor (CWV) using the surface returns. However, the proportionality given by equation 6 assumes that the absorption from other gases as well as the particulate extinction and backscattering coefficient between the two radar tones were similar which might not be true under certain hydrometeor burdens.

In this study, a range-gated radar system is simulated to explore the uncertainties in the estimates of total CWV as well as in the profiles of water vapor inside cloudy and rainy scenarios using a state of the art radar forward model. This model is not based upon equation 6 but rather a more complete version of equation 1 that also includes multiple scattering. Through this model we asses the impact of spectral variation of the particulate extinction and the backscatter coefficient, the impact due to absorption of other gases, the impact of the temperature and pressure profiles assumed, the impact of the assumed hydrometeor particle size distribution and the impact of the spectroscopy uncertainties, among others.



## 3 Radiometric Model

Radar returns were simulated using the same radiometric model as discussed in Millán et al. (2014). In short, radar reflectivities were computed using Time Dependent Two-Stream approximation (Hogan and Battaglia, 2008), assuming spherical hydrometeors, evaluating the gaseous properties using the clear sky forward model for the EOS Microwave Limb Sounder (Read et al.,

2004) and computing the surface reflection using a quasi-specular scattering model for the ocean surface. See table 1 for more details. Note that, even-though all the simulations presented in this study used an ocean backscatter model, typical land surface back scattering coefficients are also weakly frequency-dependent and hence, due to the differential nature of the technique, the results shown here can reasonably be expected to be similar to those found over land.

Throughout the study this radiometric model is run, to simulate radar tones close the 183 GHz absorption line, using CloudSat

retrievals as inputs. In particular we use the liquid water content (LWC) and ice water content (IWC) profiles from the 2B-CWC-RO R04 (Austin and Stephens, 2001; Austin et al., 2009) products, the rain and snow profiles from the 2C-RAIN-PROFILE (Lebsock and L'Ecuyer, 2011) products, and the temperature, pressure, water vapor and ozone from the European Centre for Medium-Range Weather Forecasts auxiliary (ECMWF-aux) products (Partain, 2007). The ECMWF-aux data are ECMWF weather analysis outputs interpolated in time and space to the CloudSat measurements. We also use the 2B-CLDCLASS

product (Sassen and Wang, 2008) for cloud classification. Furthermore, we assume a radar with the same similar detectable signal (the radar sensitivity) and the same radar precision as CloudSat's Cloud Profiling Radar (Tanelli et al., 2008); that is to say -30 dBZ and 0.16 dBZ, respectively. We also assume a similar vertical ranging and horizontal sampling resolution; 500 m and around 1 km, respectively.

As an example to illustrate the nature of the measurement, figure 2 shows a cross section of CloudSat-driven simulations

over the south Atlantic ocean. This cross section consists of 700 CloudSat profiles encompassing high thin cirrus, some liquid clouds, rain and snow. As shown, the water vapor field not only decreases exponentially with height, it also shows a tongue that increases with height along the track, starting from around 32$^o$S at 5 km and finishing at around 35.5$^o$S and 10 km. The impact of this water vapor burden can be seen in the simulated 170 GHz radar reflectivity subplot, at this frequency, the radar signal has been considerably more attenuated than at 94 GHz (the CloudSat radar tone, which was placed far away from any absorption

line). Furthermore, the radar reflectivity difference (177-170 GHz) already displays a resemblance to the water vapor field, for example the influence of the water vapor tongue around 10 km and 34.5$^o$S can already be seen in the radar reflectivity difference.

Figure 3 illustrates the relationship between water vapor path and radar reflectivity difference using CloudSat-driven simulations. As predicted by equation 6 there should be a linear relationship between water vapor path and the radar reflectivity

difference, either when the scattering target is the surface or hydrometeors. In the first instance, the radar reflectivity difference will be proportional to the total CWV while in the latter it will be proportional to the partial CWV, the amount of water vapor above the scattering target. As shown, even under moderately rainy scenarios, the linear relationship is clearly apparent. The slope of these curves is the exploitable signal, which is on the order of 2.3 db / 1 cm for this pair of radar tones, while the scatter may be interpreted as noise. Observe that under clear sky conditions a robust linear relationship is found between the





total CWV and the surface return difference. The spread results from the different temperature and pressure profiles for the same water vapor burden. Under cloudy and rainy scenarios the spread also increases proportionally to the mass of condensate.

The root mean square errors ($rms_e$) of these linear fits can be interpreted as the maximum likely precision error of a water vapor retrieval. The results are promising. They show that even without knowledge of the temperature, pressure, and hydrometeor

burden partial and total CWV can be constrained to less than 0.5 cm under clear and cloudy scenarios, and to under 0.7 cm for rainy ones. Note that, if the hydrometeor burden is known (for example using the range-gated information from the 160 GHz radar tone) one could subset the data using this ancillary information to constrain the partial and total CWV better.

It is important to notice that as the attenuation provides the differential radar signal it will also limit the penetration depth, in other words, it will also determine which altitude range of the atmosphere can be sampled. Figure 3 used the 170–160 GHz

frequency pair suggested by Lebsock et al. (2015) to best maximize the signal for vapor profiling within the boundary layer. These frequencies were chosen in the wing of the 183 GHz absorption line to be able penetrate the large water vapor concentrations residing in the boundary layer. However, to maximize the signal at higher altitudes the required frequency pairs will need to be selected closer to the line center, despite not being able to penetrate all the way to the surface. Figure 4 shows the penetration percentage for frequencies used throughout this study computed using around one and a half million CloudSat

pixels (the first 10 days of January 2007) and assuming a sensitivity of -30dBZ. As shown, frequencies higher than 177 GHz are severely impacted by attenuation in the lowest few kilometers of the atmosphere. However, notice that the strong surface reflection is detectable 60% of the time even at 178 GHz. The strength of the surface reflection should enable retrievals of total CWV in a diverse range of environments even where profiling is not possible.

## 4   Total Column Water Vapor Results

To properly explore the capability of this technique to estimate total CWV we have performed end-to-end retrievals. The retrieval algorithm used was a linear least square fit which allow us to quantify both the expected precision and the systematic errors of the total CWV estimates using ancillary knowledge of temperature, pressure and the hydrometeor profile. This retrieval does not use any *a-priori* information, that is to say, we do not use any additional information to constrain there retrievals. The expected precision is determined by the random noise in the radar measurements propagated through the retrieval

algorithm while the systematic errors will arise from the uncertainties in the ancillary knowledge used, as well as from the spectroscopy uncertainties.

For a given scene, a perturbed set of radar measurements is generated for each systematic uncertainty and run through the retrieval algorithm. Each of the retrieved results is then compared to the retrieved total CWV for an unperturbed run to estimate the impact of such perturbation. A list of the perturbations used can be found in table 2. Figure 5 (middle column) shows the

systematic error characterization for 3 different scenarios using the 160 and 170 GHz radar tones.

Employing this end-to-end retrieval framework allows us to search for an optimum pair of radar tones that will minimize the total error. For each scene, we simply run end-to-end retrievals to explore a given frequency space and then we choose the best radar tone combination. To find globally optimized radar tones, we performed around thirty end-to-end retrievals for each scene





type (clear sky, cirrus, altostratus, altocumulus, stratocumulus, cumulus, nimbostratus, drizzle, slight rain and moderate rain) using all the possible frequencies combination between 160 and 183 GHz, every 1 GHz step. These scenes were constructed from CloudSat representative profiles spread through-out 2007. The optimum radar tones are found in table 3. To ensure that the radar tones can be use globally we only use radar tones that were able to penetrate all the way to the surface at least 80%

of the time for clear-sky and cloudy scenes and at least 60% for rainy ones. For clear sky cases the chosen tones are separated as much as possible (14 GHz baseline) to maximize the signal (i.e the attenuation in the "on" channel) and hence minimize the impact of the random noise. For rainy cases, the radar tones are close to each other (just 3 GHz baseline) to minimize the impact of the uncertainties associated with the hydrometeors. Lastly, the cloudy cases sits in between these two extremes (8 GHz baseline), trying to minimize both, the precision and the systematic errors.

Note that these tones are optimal in the sense that they minimize the overall error, however there may be pairs of radar tones better suited for individual cases. Figure 5 (right column) shows impact of using the optimized radar tones. Overall, the precision has improved and the maximum potential biases (the root-sum-square combination of all the systematic error sources) have decreased compared to the results using the 160 and 170 GHz radar tones. Observe that for the drizzle case, even-though the precision has decreased, the optimum radar tones minimize the total error. In general, the most persistent potential bias is

due to the water vapor line width uncertainty, followed by the assumed pressure profile. In cloudy and rainy situations this is followed by the hydrometeor error as well as their corresponding PSD uncertainties. Lastly, biases induced by uncertainties in temperature, surface wind, the background atmospheric absorption from $O_2$, $N_2$ and $H_2O$ (i.e. the absorption continuum), as well as the water vapor line strength, are negligible.

To fully test the optimized radar tones, we have run end-to-end retrieval simulations for CloudSat measurements from 15 of

January 2007 (more than 150,000 pixels). Histograms and cumulative histograms of the maximum potential biases are shown in figure 6. Even under rainy conditions the expected precision was found to be on average 0.03 cm with potential biases smaller than 0.26 cm 80% of the time. Table 3 lists the precision and potential biases for all weather conditions. This expected precision is half of the Advanced Microwave Scanning Radiometer (AMSR) expected total CWV precision reported by (Wentz and Meissner, 2000). The greater precision of DAR relative to the passive microwave results from the fact that DAR makes use

of the stronger 183 GHz line whereas passive microwave relies on the 22 GHz line.

Arguably to date, passive microwave instruments have provided the benchmark for CWV measurements.Not only might DAR total CWV have better precision but could also have a considerable better horizontal resolution, i.e. around 1 km rather than the native passive microwave footprint of ∼24 km. Further, DAR total CWV estimates will be available over land and ocean rather that just over the oceans.

**5   Profiling Capabilities**

As with total CWV, we have used end-to-end retrievals to further study the capabilities of this technique to estimate profiles of water vapor under cloudy and rainy scenarios. In this case, we have used an optimal estimation algorithm (Rodgers, 2000). This algorithm uses *a priori* data to constrain the retrieval. This additional information acts as an extra set of measurements





and the solution can be thought as a weighted average between the measurements and the *a priori*. The *a priori* used is the mean profile of 100 adjacent CloudSat ECMWF-aux water vapor values smoothed by with a boxcar average of 4 vertical levels. The uncertainties in this *a priori* are assumed to be 100% allowing the information to arise mostly from the simulated measurements.

Figure 7 exemplifies the profiling capabilities of this technique for a raining profile. As shown, different pairs of radar tones can sample different parts of the cloudy/rainy atmosphere: tones close to the line center can sample higher altitudes while tones with moderate water vapor absorption can penetrate further into the surface. These can be appreciated in the averaging kernels subplots. These kernels delimit the region of the atmosphere from which the atmospheric information is contributing to the retrieved values at a given altitude (Rodgers, 2000). Hence, if only two radar tones are going to be used, they will need

to be carefully chosen to be able to sample the desired vertical region. In this study, altitudes where the kernel maximum is greater than 0.4 are considered to have retrievals not influenced too much by the *a priori* and therefore carry useful retrieved information. The expected precision and systematic errors are only shown for those altitudes. As shown in the last row of figure 7, using multiple radar tones it is possible to sample the entire vertical profile. Figure 8 shows the retrieved water vapor profiles over the same cross section displayed in Figure 2 C. As shown, this technique could provide valuable information for

studies of water vapor vertical and horizontal distribution in cloudy/rainy areas and as input to weather forecasting models, complementing well the existing water vapor measurements. The use of multiple radar tones is here analyzed purely from an information content point of view with no concern for practical implementation matters.

Note that the tone pairs used in figure 7 and figure 8 are optimized for different vertical regions (except for the multi-tones retrievals). These radar tones were found using the same approach as for total CWV vapor. These optimum radar tones can

be found in table 3. If only two tones are available, to ensure that they can penetrate down to the lowest desired altitude, radar tones better suited for higher altitudes need to be sacrificed. As before, to test the applicability of these radar tones, we run end-to-end retrieval simulations for CloudSat measurements from the 15th of January 2007. Histograms and cumulative histograms of the maximum potential biases are shown in figure 9. The expected precision was found to be on average around 89% with a potential biases smaller than ∼80% when the profile is being retrieved close to surface but smaller than 37% above

6 km 80% of the time. At all altitude ranges, the main source of potential biases are the hydrometeor uncertainties, followed by pressure, temperature and spectroscopic uncertainties. The last three contribute at most around 10%. We also notice that an *a priori* that is too dissimilar to the atmospheric profile has the potential to be a source of systematic uncertainty impacting more at higher altitudes.

At first glance, the DAR expected precision may seem large in comparison to uncertainties of current water vapor profilers.

For example in the upper troposphere, the Microwave Limb Sounder (MLS) (Waters et al., 2006) expected precision varies from 65% to 15% (Livesey et al., 2015) while the Atmospheric Infrared Sounder (AIRS) (Aumann et al., 2003) has uncertainty estimates of around 20% (Susskind et al., 2003). However, MLS has a ∼1.5 to 3.0 km vertical resolution and a horizontal resolution of 7 km across track and 180 km along track (Livesey et al., 2015) while AIRS has a vertical resolution of around 2.7 km close to the surface and 4.3 km near the tropopause (Maddy and Barnet, 2008) with a field of regard of ∼40 km. The

potential resolution of DAR greatly exceeds the resolution of current water vapor profilers and, hence, the expected precision





needs to be interpreted accordingly. For example, simply by matching their vertical resolutions, the DAR precision could be improved by a factor of $\sim \sqrt{3}$ (when matching the 1.3 km MLS resolution) to $\sim \sqrt{8}$ (when matching the $\sim$4 km AIRS resolution) while still retaining the $\sim$1 km horizontal resolution.

## 6  Conclusions

We have discussed the theoretical capabilities of a differential absorption radar method to retrieve total column water vapor under clear sky, cloudy and precipitating conditions, as well as, water vapor profiles under cloudy and rainy conditions. This concept relies on radar reflectivities at two frequencies in the wings of the 183 GHz water vapor line ("on" and "off" an absorption line) to estimate the absorbing gas path between the radar and the scatterer.

An inversion scheme was implemented focusing on the retrieval propagation of measurement noise as well as systematic biases. This scheme provided a mathematical basis for the weighting of the water vapor signal against errors introduced by uncertainties in other parameters needed by the retrieval such as the assumed pressure and temperature vertical distribution, hydrometeor abundances, particle size distributions, as well as spectroscopic uncertainties. Then, this scheme was used to select pair of radar tones that maximize the water vapor signal and minimize the total error at different targeted altitude ranges. As expected, to sound close to the surface, the inversion scheme selected radar tones well into the the wing of the 183 GHz to be able penetrate the large water vapor concentrations residing in the boundary layer; and, radar tones closer to the line center to sound higher altitude ranges.

Assuming an instrument precision of 0.16 dBZ and a radar sensitivity of -30dBZ and a retrieved vertical resolution of 500 m against a $\sim$1.7 km footprint, we found that even under rainy conditions the total column water vapor expected precision will be on average 0.03 cm with potential biases smaller than 0.26 cm 80% of the time. This precision is half as good precision of passive microwave total column water vapor measurements with the potential of considerably better horizontal resolution. Further, DAR total column water vapor estimates would be available over land and ocean and essentially all-sky because, for the radar tones selected, the surface return is always above the radar sensitivity limit.

For water vapor profiles, the expected precision was found to be on average around 89% with a potential biases 80% of the time smaller than 77% when the profile is being retrieved close to surface but smaller than 38% above 3 km. Simply by matching the vertical resolution of current humidity sounders the DAR precision could be improved considerably still retaining the high horizontal resolution. Furthermore, DAR specifically samples in cloud and rain where existing sensors suffer large errors or simply cannot measure. At all altitude ranges, the main source of potential biases are the hydrometeor uncertainties and any attempt to development an instrument should be probably coupled with an effort to characterized the hydrometeor particle size distributions better. These results demonstrate that this technique holds considerable potential as a method for retrieving water vapor profiles under realistic cloudy and precipitating scenarios.

*Acknowledgements.* The research described in this paper was carried out by the Jet Propulsion Laboratory, California Institute of Technology, under contract with the National Aeronautics and Space Administration.



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




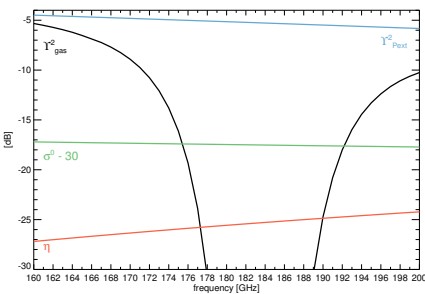

**Figure 1.** Example of a typical atmospheric transmittance due to gases ($\Upsilon^2_{gas}(\nu)$, black) and hydrometeors ($\Upsilon^2_{Pext}(\nu)$, blue), as well as the backscatter reflectivity ($\eta(\nu)$, red), and the ocean backscatter ($\sigma^2_0(\nu)$ green, offsetted by 30dB) for a surface return journey (downward atmospheric pass, surface reflection, upward pass) for a nimbostratus cloud near the 183 GHz $H_2O$ band region. The atmospheric profile was taken from CloudSat data as described in section 3. The ocean backscatter corresponds to a surface wind of $3\,\mathrm{m\,s^{-1}}$ and temperature of $28\,^\circ$C. Note that only the transmittance due to gases show a significant frequency dependence.

**Table 1.** Radar Model Specifics

| Parameter | Detail |
| --- | --- |
| Water Dielectric Properties | Liebe et al. (1991) |
| Ice Dielectric Properties | Hufford (1991) |
| Ice Water Content (IWC) PSD[a] | McFarquhar and Heymsfield (1997) |
| Liquid Water Content (LWC) PSD | Using a log normal distribution with a 10 μm mean radius and a 1.3 spread. |
| Rain PSD | Abel and Boutle (2012) |
| Snow PSD | Sekhon and Srivastava (1970) |
| Gas Absorption | Read et al. (2004) |
| Radiation Propagation | Hogan and Battaglia (2008); Hogan (2013) |
| Surface Reflection | Li et al. (2005b) assuming a surface wind of $3\,\mathrm{m\,s^{-1}}$, sea surface temperature of $28^o$C, a Fresnel fraction of 1 and zero salinity. |

[a] particle size distribution.

[b] only applied to the water vapor profile retrievals.



**Table 2.** Systematic uncertainties perturbations[a].

| Perturbation | Amount | Comments |
| --- | --- | --- |
| Temperature | 3 K | Calculated as the average of 10 randomly perturbed profiles |
| Pressure | 5 % | Calculated as the average of 10 randomly perturbed profiles |
| IWC error | 50 % | – |
| LWC error | 50 % | – |
| Rain error | 50 % | – |
| Snow error | 50 % | – |
| IWC PSD[b]1 | – | Heymsfield et al. (2002) |
| IWC PSD2 | – | Donovan and van Lammeren (2002) |
| LWC PSD1 | – | Lognormal distribution with a 6 μm mean radius and a 1.5 spread. |
| Rain PSD1 | – | Marshall and Palmer (1948) |
| Rain PSD2 | – | Willis (1984) |
| Snow PSD1 | – | Gunn and Marshall (1958) |
| Surface Wind | 12 m s$^{-1}$ | – |
| Line Strength | 0.25% | Pickett (1998) |
| Line Width | 4% | Bauer et al. (1989), Goyette and DeLucia (1990) |
| $H_2O$ Continuum | 10% | Meshkov (2006) |
| $N_2$ and $O_2$ Continuum | 10% | Meshkov (2006) |
| *A priori*[c] | 20% | Calculated as the average of 10 randomly perturbed profiles |

[a] for the "unperturbed" characteristics see table 1.

[b] particle size distribution

[c] only applied to profile retrievals.





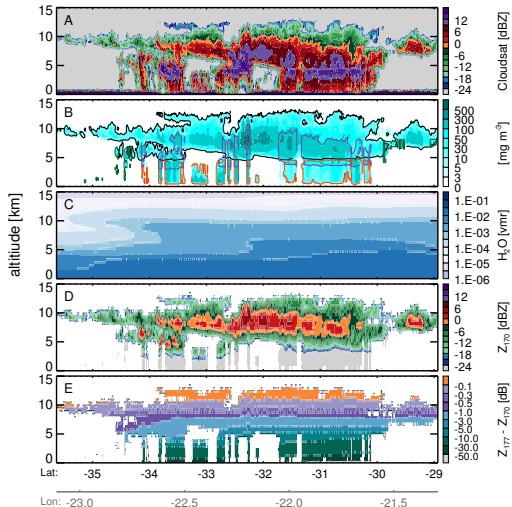

**Figure 2.** Cross section exemplyfing the CloudSat-driven simulations (data from January 15, 2007 over the South Atlantic ocean). (A) Cloudsat radar reflectivity. (B) CloudSat retrieved total (IWC+LWC+Rain+Snow) hydrometeor water content. Black, green, red and purple lines, respectively, delimit areas where IWC, LWC, rain and snow were present. (C) ECMWF-aux water vapor. (D) Simulated CloudSat-driven radar reflectivity at 170 GHz. (D) Simulated radar reflectivity difference (177-170 GHz).

**Table 3.** Optimum radar tones

|  | Scene type | Radar Tones [GHz] | Precision[a] | Potential bias[a] 60 % of the time | Potential bias[a] 80 % of the time |
|---|---|---|---|---|---|
| Total CWV [cm] | Clear Sky | 160, 174 | $0.01 \pm 0.001$ | 0.10 | 0.14 |
|  | Cloudy | 166, 174 | $0.01 \pm 0.001$ | 0.10 | 0.14 |
|  | Rainy[b] | 169, 172 | $0.03 \pm 0.008$ | 0.18 | 0.26 |
| Profile $H_2O$ [%] | <9 km | 178, 183 | $86 \pm 33$ | 20 | 26 |
|  | 9-12 km | 178, 183 | $84 \pm 77$ | 13 | 20 |
|  | 6-9 km | 178, 181 | $93 \pm 83$ | 25 | 37 |
|  | 3-6 km | 169, 177 | $69 \pm 63$ | 48 | 81 |
|  | 1-3 km | 162, 172 | $87 \pm 64$ | 50 | 78 |
|  | 0-1 km | 160, 170 | $114 \pm 77$ | 29 | 70 |

[a] Estimates computed using end-to-end retrievals for each of the CloudSat measurements available in 15 January 2007.

[b] rain rates lower $10 \, \mathrm{mm \, h^{-1}}$.





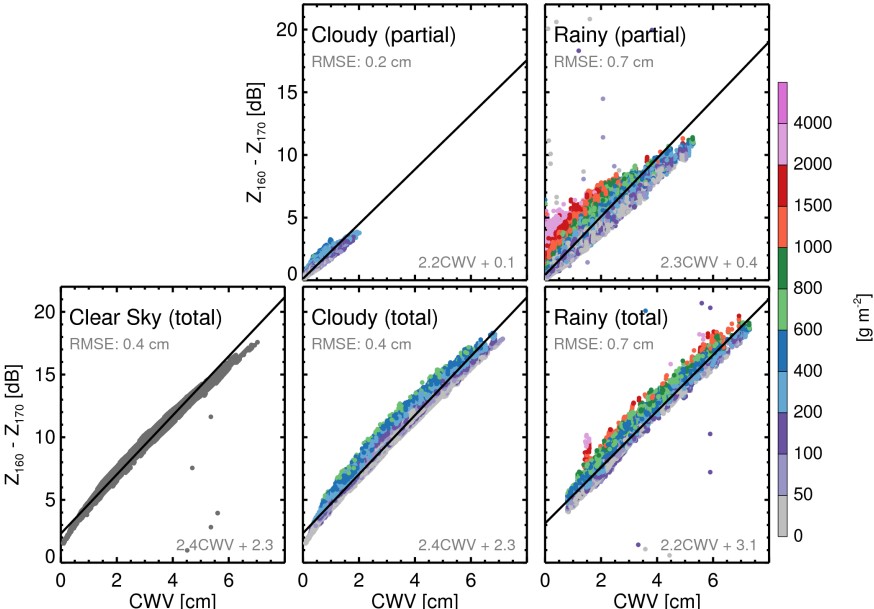

**Figure 3.** Simulated relationships between surface radar returns and total and partial CWV. Each point represents a CloudSat-driven simulation for each of the CloudSat measurements available in 15 January 2007. The total or partial hydrometeor column is color coded. Dark gray is used for clear sky cases (total hydrometeor column equal to zero). The black line shows the linear regression. The root mean square error (rms$_e$) displayed is the overall linear regression error for each scenario. The top row shows the relationship between the range-gated radar returns and partial CWV and the bottom row displays the relationship between the surface returns and total CWV.

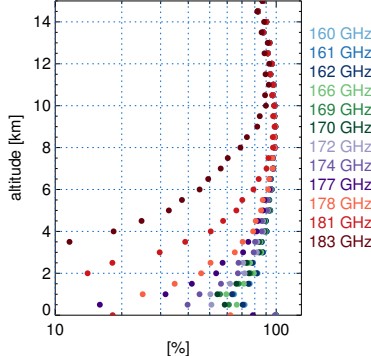

**Figure 4.** Percentage penetration for frequencies discussed in this study calculated using CloudSat-driven simulations for ten days assuming a radar sensitivity of -30 dBZ.



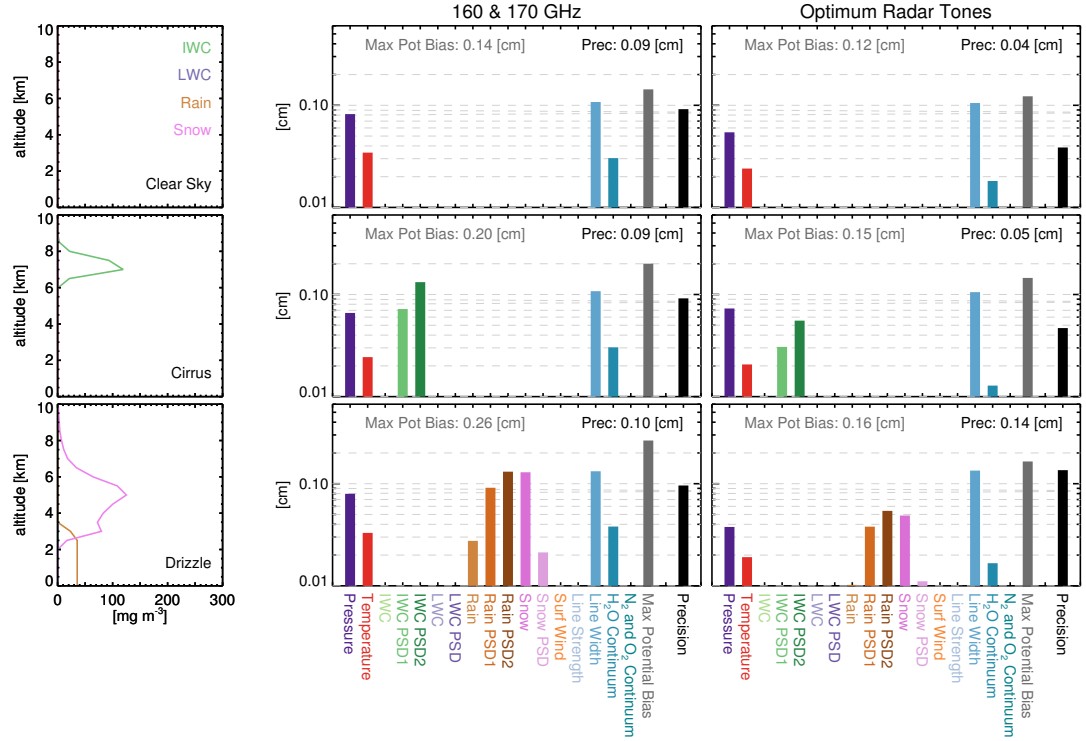

**Figure 5.** Systematic error estimates caused by each of the sources described in Table 2 as well as the precision and maximum potential bias for three different scenarios (Clear sky, Cirrus,and drizzle). The maximum potential bias is the root-sum-square combination of all the systematic error sources. (left) hydrometeor burden for each of the scenarios, (middle) simulations performed using 160 and 170 GHz radar tones, and (right) simulation using optimum radar tones (see table 3).

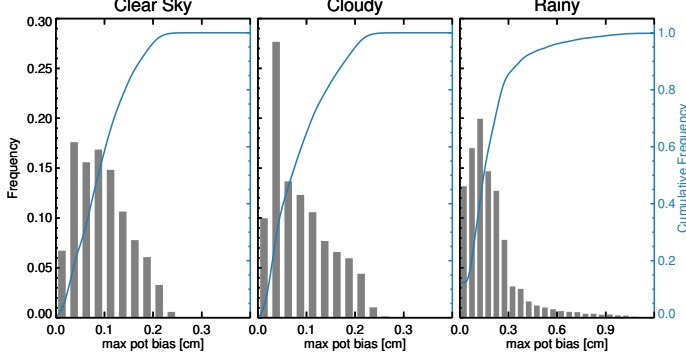

**Figure 6.** Histogram (gray) and cumulative histogram (blue) of the maximum potential biases for CloudSat-driven end-to-end total column water vapor retrievals for each of the CloudSat measurements available in 15 January 2007.



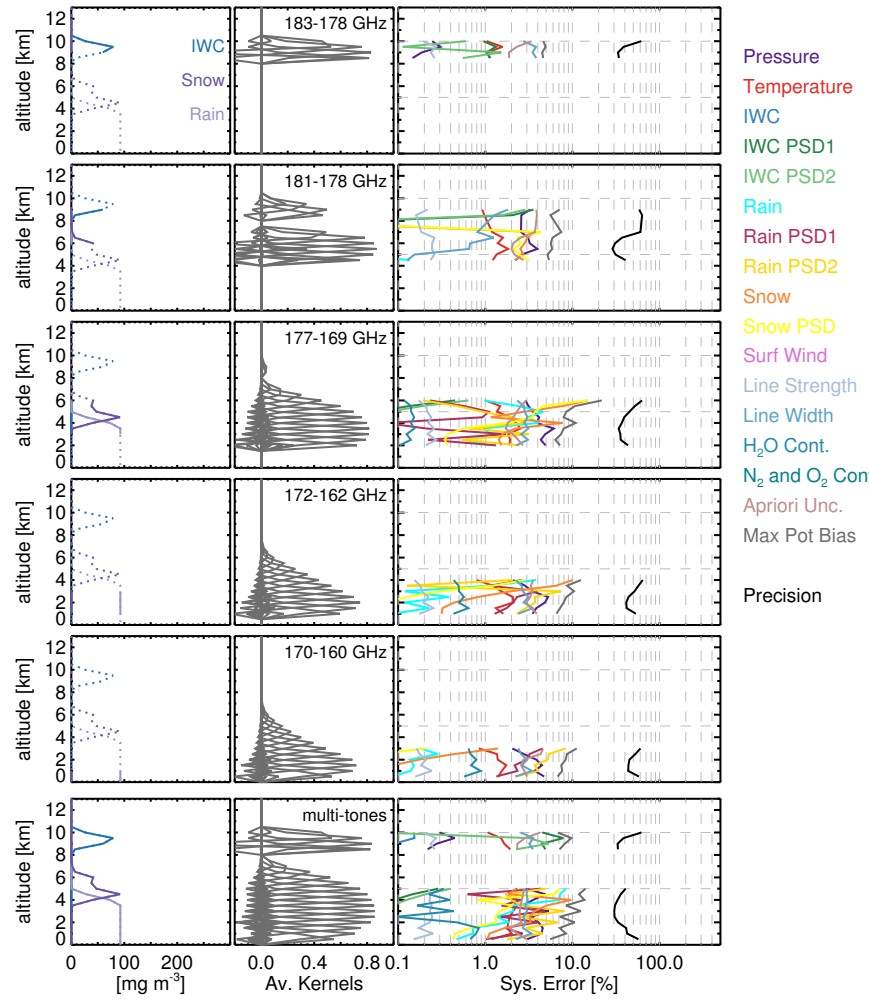

**Figure 7.** (left) Color coded hydrometeor burden. Dotted lines show the entire profile while solid lines display the vertical range for which the radar tones used are better suited. (middle) Averaging kernels. (right) Systematic error estimates caused by each of the sources described in Table 2 as well as the precision and maximum potential bias. These errors are only shown for altitudes with kernels greater than 0.4 which indicates retrievals not influenced by the *a priori*. The top five rows show "optimized" radar tones for different vertical regions. The last row displays an example of how multi-tones (183, 177, 170 and 160 GHz frequencies) retrievals can be use to sample an ample altitude range. Note that these multi-tone frequencies are not optimized.





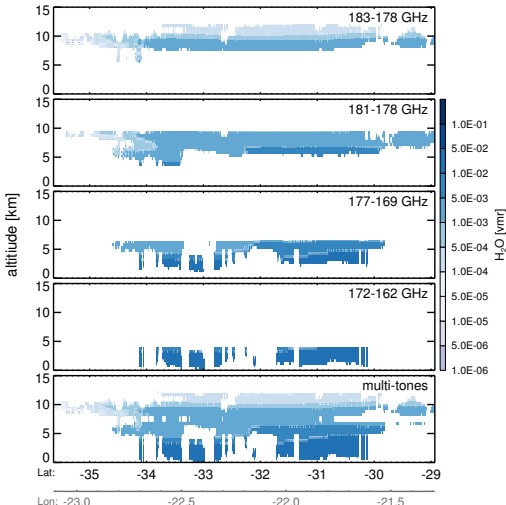

**Figure 8.** Cross section of CloudSat-driven water vapor retrievals retrievals. This is the same cross section as in figure 2C. Water vapor values are shown only for altitudes with kernels greater than 0.4 which indicates retrievals not influenced by the *a priori*. The retrieved values using the 170-160 radar tone combination are not displayed because, as shown in figure 2B, there were no hydrometeors close to the surface.

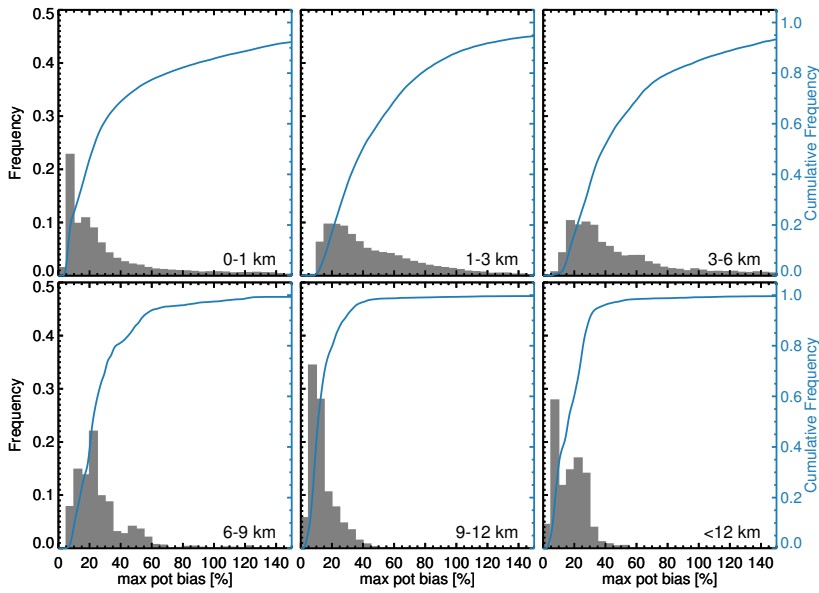

**Figure 9.** Histogram (gray) and cumulative histogram (blue) of the maximum potential biases for CloudSat- driven end-to-end profile water vapor retrievals for each of the CloudSat measurements available in January 15, 2007.