# Peer review of "Differential Absorption Radar Techniques: Water Vapor Retrievals"

_Atmospheric Measurement Techniques, 2016_

## Referee Comment (RC1) · Anonymous Referee #1 · 6 Apr 2016

The paper is an extension of work by Lebsock et al (2015). In the present paper, a much more detailed error model is used to evaluate the method. The authors also consider different wavelength combinations to optimize the retrievals under clear, cloudy and rainy conditions. I found the paper to be informative and well written. I was interested to find that the 'most persistent potential bias is due to the water vapor line width uncertainty' (p. 6).

I recommend publication. There are, however, several issues below that the authors should address.

For the Ku- and Ka-band DPR radar aboard the GPM satellite, the standard deviation of the normalized surface cross section, NRCS, is quite high over land at nadir and near-nadir incidence at both frequencies – greater than 5 dB in many cases. Over ocean

[Figure]

and off-nadir incidence over land, the standard deviation is much smaller, usually on the order of 1-2 dB. I'm less familiar with this variability for CloudSat but the authors should know these data well; I'm surprised not to see this as part of the uncertainties listed in Table 2.

On the other hand, since the measurement is differential, I would expect the errors caused by variations in the NRCS to be much less since the quantity of interest is the variance of the difference rather than the variance of either NRCS alone. This suggests that the quantities that need to be specified are the variances of the NRCS and the correlation coefficient, p, of the NRCS at the two frequencies. Although I would expect p to be close to 1, this value, as well as the variance, will be a function of incidence angle, surface type and frequency separation and could be important parameters to be considered in the radar design.

For range profiling, the analogous assumption to constant or known variation in the NRCS is that the radar reflectivity factors be constant at the two frequencies (Z(f1)=Z(f2)). Although for most clouds, this assumption is reasonable, under raining conditions, I would guess that the assumption is problematic and that this is the main reason for choosing the frequencies to be close together (169, 172 GHz). Is this correct? A plot of the difference Z(f1)-Z(f2) versus f2-f1 (for a center frequency of, say, 170 GHz), using one of the rain or snow PSD's in the table, would be useful.

Is the focus around 183 GHz rather than around 22 GHz because of the larger dynamic range available at 183 GHz or is it because of an interest in cloud profiling rather than rain profiling? Or is it because there are more cloudy regions than raining regions? Although the issues of cloud detection and estimation are not discussed, I assume that the objective is to retrieve cloud parameters as well as water vapor.

p. 2, line 27: 'dielectric factor' rather than 'dielectric constant' – dielectric factor is a function of the dielectric constant but not identical to it.

p. 4, line 9: 'close to the 183 GHz . . .'

p. 5, lines 17-18: 'The strength of the surface reflection ..' But this strength depends on surface type and incidence angle that will affect the dynamic range.

p. 6, line 4: 'used' rather than 'use'.

p. 8, line 28: 'to develop'; 'to characterize'.

———————————————————

---

## Author Comment (AC1) · 16 Apr 2016

We thank the reviewer for his/her comments. Below are our responses in blue.

The paper is an extension of work by Lebsock et al (2015). In the present paper, a much more detailed error model is used to evaluate the method. The authors also consider different wavelength combinations to optimize the retrievals under clear, cloudy and rainy conditions. I found the paper to be informative and well written. I was interested to find that the 'most persistent potential bias is due to the water vapor line width uncertainty' (p. 6).

I recommend publication. There are, however, several issues below that the authors should address.

[Figure]

For the Ku- and Ka-band DPR radar aboard the GPM satellite, the standard deviation of the normalized surface cross section, NRCS, is quite high over land at nadir and near-nadir incidence at both frequencies – greater than 5 dB in many cases. Over ocean and off-nadir incidence over land, the standard deviation is much smaller, usually on the order of 1-2 dB. I'm less familiar with this variability for CloudSat but the authors should know these data well; I'm surprised not to see this as part of the uncertainties listed in Table 2.

On the other hand, since the measurement is differential, I would expect the errors caused by variations in the NRCS to be much less since the quantity of interest is the variance of the difference rather than the variance of either NRCS alone. This suggests that the quantities that need to be specified are the variances of the NRCS and the correlation coefficient, p, of the NRCS at the two frequencies. Although I would expect p to be close to 1, this value, as well as the variance, will be a function of incidence angle, surface type and frequency separation and could be important parameters to be considered in the radar design.

Similar land ocean differences in the NRCS standard deviation are observed in CloudSat data, and we expect similar variability in G-BAND NRCS. We characterize the NRCS uncertainty by the uncertainty called Surface Wind in Table 2. The change in surface wind affects the surface roughness, which in turn changes the NRCS. (We assumed a surface wind of 12 ms$^{-1}$ rather than 3 ms$^{-1}$ (see table 1.)). The reviewer is correct in assume that errors caused by variation in the NRCS are tiny due to the differential nature of the measurement. As shown in Figure 5, the uncertainty due to Surface wind is less than 0.01 cm. This error may be slightly larger over land surfaces, however we do not have an adequate model to test this behavior and the differential nature of the measurement provides good reason to suspect that this error will remain

small. This is discussed in the first paragraph of Radiometric Model section. In table 2 we will add in the comment section for surface wind: "to characterize uncertainties in $\sigma_0(\nu)$"

With respect to the incidence angle we will specify in section 3, that we are only assuming Nadir viewing angles in this study. Further we will add a sentence stating: "Viewing angles off the nadir, provided by using a scanning radar, are not explored in this study but, apart from the extra attenuation due to the longer paths, are fundamentally the same as when using the nadir view."

For range profiling, the analogous assumption to constant or known variation in the NRCS is that the radar reflectivity factors be constant at the two frequencies (Z(f1)=Z(f2)). Although for most clouds, this assumption is reasonable, under raining conditions, I would guess that the assumption is problematic and that this is the main reason for choosing the frequencies to be close together (169, 172 GHz). Is this correct? A plot of the difference Z(f1)-Z(f2) versus f2-f1 (for a center frequency of, say, 170 GHz), using one of the rain or snow PSD's in the table, would be useful.

That is correct. As is stated in page 6 line 7 of the original document for total column water measurements. Further, that is why for range profiling (page 7 line 25) the main uncertainty are the hydrometeors uncertainties. Even though, the optimum frequency selection process attempted to minimize them they still dominate the error budget.
We will update Figure 1 to include perturbed transmittance and backscattering reflectivites due to hydrometeors, as well as the ocean. See updated figure attached. The caption will be modified to include: "Thin line show the impact of assuming a different particle size distribution or a different surface wind. These lines have been offseted to ease comparison against the unperturbed ones." Further, at the end of the theoretical basis section, after: "Through this model we asses the impact of spectral variation of

the particulate extinction and the backscatter coefficient, the impact due to absorption of other gases, the impact of the temperature and pressure profiles assumed, the impact of the assumed hydrometeor particle size distribution and the impact of the spectroscopy uncertainties, among others" we will add: "(see for example, Figure 1 thin lines)"

Is the focus around 183 GHz rather than around 22 GHz because of the larger dynamic range available at 183 GHz or is it because of an interest in cloud profiling rather than rain profiling? Or is it because there are more cloudy regions than raining regions?
In general, it is due to the larger dynamical range available at 183 GHz but the reviewer is correct in pointing out that there are far more cloudy targets than rainy targets. In the introduction we will add the following sentence: The water vapor line at 183 GHz is used rather than the 22 GHz because its attenuation is stronger which provides a greater dynamical range allowing us to explore cloud and rain profiling.

Although the issues of cloud detection and estimation are not discussed, I assume that the objective is to retrieve cloud parameters as well as water vapor.
Yes, with an off-line radar tone. At the end of the first paragraph in section 4, we will add something along the lines of "These end-to-end retrievals assume knowledge of the hydrometeors vertical distribution. This knowledge is assumed to come from an off-line radar tone using CloudSat-like retrievals. The impact of attenuation on this radar tone/retrievals is not investigated here."

p. 2, line 27: 'dielectric factor' rather than 'dielectric constant' – dielectric factor is a function of the dielectric constant but not identical to it. In this case, we do use the dielectric factor of the target. It is not a constant in our radiometric model.

p. 4, line 9: 'close to the 183 GHz ...' ok

p. 5, lines 17-18: 'The strength of the surface reflection ..' But this strength depends on surface type and incidence angle that will affect the dynamic range.
That is correct but as stated on page 4 line 6 of the original document (section 3 - Radiometric model): Note that, even-though all the simulations presented in this study used an ocean backscatter model, typical land surface back scattering coefficients are also weakly frequency-dependent and hence, due to the differential nature of the technique, the results shown here can reasonably be expected to be similar to those found over land.

With respect of the viewing angles: nadir viewing is assumed throughout the study, see response above

p. 6, line 4: 'used' rather than 'use'. ok

p. 8, line 28: 'to develop'; 'to characterize'. ok

―――――――――――――

[Figure]

Fig. 1.

---

## Referee Comment (RC2) · Anonymous Referee #2 · 27 Apr 2016

**Review of amt-2016-72**

|        |                                                              |
|--------|--------------------------------------------------------------|
| Title: | A Differential Absorption Radar Techniques: Water Vapor Retrievals |
| Authors: | Millán et al.                                              |

**1 Summary**

In this work, water vapor retrieval from satellite-based radar measurements was first described and further demonstrated and investigated with simulations. The ideal is to exploit the differential water absorption from two distinct radar frequencies at on and off the absorption line (183 GHz). As a result, the total column water vapor as well as water vapor profile can be estimated. The feasibility of the retrieval was investigated under different weather conditions including clear sky, cloudy, and rainy. It is also very valuable to assess the performance of different frequency combinations and contributions from various error sources. In summary, this is a well organized and written paper with significant scientific contribution to the community. I recommend to publish this work after those relatively minor comments/concerns in the following two sections are addressed.

**2 General Comments**

As pointed out by the authors, it is important to measure water vapor with adequate resolution, accuracy, and coverage to characterize the atmosphere. The proposed satellite-based retrieval method exploits differential absorption at two radar frequencies at around water vapor absorption. The overall structure of the paper is well organized and clear. However, there are some parts of the paper can be explained in a more clear way. For example, the theoretical basis is easy to understand. However, the forward model for simulation is not clearly explained. Even though the reference paper provided does not have enough detail to understand the basic idea of the simulation. I suggest to include a flow chart outlining the important steps in the simulation, which can be provided in the appendix. Specifically, what are the outputs of the simulations, reflectivity at two assigned frequencies? How are the reflectivity from each gate generated? When the noise is added, is it added in the dB scale or linear scale? Additionally, it is not clear to how the sensitivity of -30 dBZ, instrument precision of 0.16 dBZ, and spatial resolution were incorporated in the simulations, or were they ever incorporated in the simulation?

Moreover, the retrieval of total CWV and profile of water vapor needs to be elaborated. For example, in line 21 page 5, it is stated that "The retrieval algorithm used was a linear least square fit ..." It is not clear what variables the least square fit are applied to. Do you mean reflectivity from multiple frequencies? In the profiling case, more detail for the optimal estimation algorithm is desired. Do you mean the problem is postulated as a constraint optimization? What is the mathematical representation of it and how exactly are the measurement and a priori information used in this method?

**3  Specific Comments**

The comments here are relatively minor and mostly editorial.

1. *Line 1, page 2 and line 27, page 3*: change "asses" to "assess".

2. *Line 3, page 2*: change "retrieved" to "retrieve"

3. *equation (6)*: need $dr$

4. *Line 15, page 3*: "same similar"?

5. *Line 23, page 5*: change "there" to "these"

6. *Line 26, page 5*: need a space before "Not only ..."

7. *Line 8, page 7*: What is the role of these kernels? Please elaborate.

---

## Short Comment (SC1) · 17 May 2016

These results are fairly limited because obviously the authors seem to omit the analysis of noise errors in the derivation of water-vapor profiles in clouds. The DAR ist extremely sensitive to the SNR of the online and offline signals because the relative error in the derivation of water vapor is inversely proportional to the differential optical thickness. There is a vast of literature from the investigation of water-vapor differential absorption lidar, which is nearly equivalent to DAR but is it not mentioned by the authors. I expect that it will turn out that the SNR of the return signals is far away from enabling reasonable measurements of water vapor in clouds from space. Before this analysis is done and this ommission is healed, it is hardly possible to make a reasonable judgement of this technique. I strongly recommend that the methodological analysis of DAR is extended by system noise error propagation. Therefore, also the argument

that the accuracy of the measurement is increased by averaging is incorrect because the authors are dealing with systematic errors but not with uncorrelated noise. In this connection, I am wondering how useful a precision of 89% is (or an error of a factor of 2). In this case, it is probably better to guess the water vapor content of the cloud by the temperature profile. An NWP output will likely produce more accurate results.

---

## Author Comment (AC2) · 20 May 2016

We thank the reviewer for his/her comments. Below are our responses in blue.

**1   Reviewer 2 comments**

**Summary**

In this work, water vapor retrieval from satellite-based radar measurements was first described and further demonstrated and investigated with simulations.  The ideal is to exploit the differential water absorption from two distinct radar frequencies at

on and off the absorption line (183 GHz). As a result, the total column water vapor as well as water vapor profile can be estimated. The feasibility of the retrieval was investigated under different weather conditions including clear sky, cloudy, and rainy. It is also very valuable to assess the performance of different frequency combinations and contributions from various error sources. In summary, this is a well organized and written paper with significant scientific contribution to the community. I recommend to publish this work after those relatively minor comments/concerns in the following two sections are addressed.

**General Comments**

As pointed out by the authors, it is important to measure water vapor with adequate resolution, accuracy, and coverage to characterize the atmosphere. The proposed satellite-based retrieval method exploits differential absorption at two radar frequencies at around water vapor absorption. The overall structure of the paper is well organized and clear. However, there are some parts of the paper can be explained in a more clear way. For example, the theoretical basis is easy to understand. However, the forward model for simulation is not clearly explained. Even though the reference paper provided does not have enough detail to understand the basic idea of the simulation. I suggest to include a flow chart outlining the important steps in the simulation, which can be provided in the appendix. Specifically, what are the outputs of the simulations, reflectivity at two assigned frequencies? How are the reflectivity from each gate generated? When the noise is added, is it added in the dB scale or linear scale? Additionally, it is not clear to how the sensitivity of -30 dBZ, instrument precision of 0.16 dBZ, and spatial resolution were incorporated in the simulations, or were they ever incorporated in the simulation?

We consider adding a flowchart but we decided against it. We believe that the second paragraph of the radiometric model in conjunction with table 1, is sufficient. Besides

reviewer 1 did not found this to be problematic. The incorporation of the noise and the radar sensitivity will be discussed in an appendix describing the retrievals, see below.

Moreover, the retrieval of total CWV and profile of water vapor needs to be elaborated. For example, in line 21 page 5, it is stated that "The retrieval algorithm used was a linear least square fit ..." It is not clear what variables the least square fit are applied to. Do you mean reflectivity from multiple frequencies? In the profiling case, more detail for the optimal estimation algorithm is desired. Do you mean the problem is postulated as a constraint optimization? What is the mathematical representation of it and how exactly are the measurement and a priori information used in this method?

An appendix will be added describing the retrievals. The appendix is below.

**Specific Comments**

The comments here are relatively minor and mostly editorial.

1. Line 1, page 2 and line 27, page 3: change "asses" to "assess". Ok

2. Line 3, page 2: change "retrieved" to "retrieve" Ok

3. equation (6): need dr Correct thanks for spotting the omission.

4. Line 15, page 4: "same similar"? similar will be deleted

5. Line 23, page 5: change "there" to "these" Ok

6. Line 26, page 5: need a space before "Not only ..." Ok

7. Line 8, page 7: What is the role of these kernels? Please elaborate. This is also discuss in appendix.

**A Least-Squares**

In this study, the least-squares retrieval is used to estimate a total water vapor column, $w$. The measurement vector $\mathbf{y}$ is determined by the differences between surface radar returns at different frequencies, that is to say

$$\mathbf{y} = [dBZ(\nu_2, r_s) - dBZ(\nu_1, r_s), dBZ(\nu_3, r_s) - dBZ(\nu_1, r_s), \cdots] \tag{1}$$

For completeness, we present the theory for multiple radar tones even though in section 4 we only used a pair. The idea is to minimize the sum of the square differences — a least-squares approach— between the measurement vector $\mathbf{y}$ and the simulated measurements, given by:

$$\hat{\mathbf{y}}(\mathbf{x}) = [\mathbf{F}_{\nu_2, r_s}(\mathbf{x}, \mathbf{b}) - \mathbf{F}_{\nu_1, r_s}(\mathbf{x}, \mathbf{b}), \mathbf{F}_{\nu_3, r_s}(\mathbf{x}, \mathbf{b}) - \mathbf{F}_{\nu_1, r_s}(\mathbf{x}, \mathbf{b}), \cdots] \tag{2}$$

where $\mathbf{F}$ is the forward model described in section 3, $\mathbf{x}$ is a water vapor linearisation profile, and $\mathbf{b}$ is known as the forward model parameter and contains additional terms needed by the forward model but not being retrieved (such as profiles of temperature, pressure, ice water content, liquid water content, rain, snow, etc). In these simulations any reflectivity below the radar sensitivity is set to missing.

The solution of such system may be found by minimizing the cost function,

$$\chi^2 = [\mathbf{y} - \hat{\mathbf{y}}(\mathbf{x})]^T \mathbf{S}_y^{-1} [\mathbf{y} - \hat{\mathbf{y}}(\mathbf{x})] \tag{3}$$

where $\mathbf{S}_y$ is the matrix describing the noise covariance of the measurements.

Following Rodgers (2000), the iterative least-squares fit solution is given by,

$$w_{i+1} = w_i + \left[\mathbf{K}^T \mathbf{S}_y^{-1} \mathbf{K}\right]^{-1} \mathbf{K}^T \mathbf{S}_y^{-1} [\mathbf{y} - \hat{\mathbf{y}}(\mathbf{x})_i)] \tag{4}$$

where the total water vapor column, $w_i$, is computed by suitably integrating the vertical profile $\mathbf{x}_i$ and

$$\mathbf{K} = \frac{\partial \hat{\mathbf{y}}(\mathbf{x})}{\partial w}|_{\mathbf{x}=\mathbf{x}_i} \tag{5}$$

is the Jacobian matrix evaluated by finite differences perturbing the entire profile by 1%. Note that after each iteration $\mathbf{x}_{i+1}$ is computed following

$$\mathbf{x}_{i+1} = \frac{w_{i+1}}{w_i}\mathbf{x}_i \tag{6}$$

.

This technique estimates the uncertainties (the precision) in the retrieved total column water vapor, $w$, according to:

$$\mathbf{S}_w = \left(\mathbf{K}^T\mathbf{S}_y^{-1}\mathbf{K}\right)^{-1} \tag{7}$$

where $\mathbf{S}_w$ is the covariance matrix of the estimated total column water $w_{i+1}$.

So, to test the total column water vapor retrieval four parameters are needed: (1) the measurements $\mathbf{y}$, (2) the initial guess $x_0$, (3) the forward model parameters $\mathbf{b}$, and (4) the measurement covariance matrix $\mathbf{S}_y$. The measurements are CloudSat-driven simulations. The initial guess, that is to say the water vapor profile used in the first iteration, is a climatological water profile. The forward model parameters needed are: IWC, LWC, rain, snow, temperature and pressure. All of them were taken from the CloudSat retrieval products. The hydrometeor PSDs used were listed in table 1, which are the same ones employed to compute the synthetic measurements. Lastly, the measurement covariance matrix was assumed to be a diagonal matrix with the variances of the elements of the measurement vector. That is to say,

$$\sigma^2 = \left(\sqrt{2\delta_Z}\right)^2 \tag{8}$$

where $\delta_Z$ is the radar precision, in this study assumed to be 0.16 dBZ, and the expression within the brackets is just the addition in quadrature of the uncorrelated radar tones precision.

While finding the solution of the retrieval problem is the central part of operational retrieval algorithms, it is not the main focus of this study. This study quantifies the theoretical capabilities of such measurements, and therefore, the precision and accuracy of the solution $w$ reached by the iterative process. As already mentioned, the uncertainty in the retrieved state due to the measurement noise (the precision) is described by the diagonal elements of the covariance matrix $\mathbf{S}_w$. To compute the accuracy impacts of various sources of systematic uncertainties were investigated. These errors were estimated using end-to-end retrieval simulations. First, for each systematic error a perturbed set of measurements were generated and ran through the retrieval algorithm. These perturbed measurements were computed following,

$$\mathbf{y}' = \mathbf{F}(\mathbf{x}_T, \mathbf{b}') \tag{9}$$

where $\mathbf{x}_T$ is the true water vapor state as provided by the CloudSat-ECMWF product, and where $\mathbf{b}'$ is the perturbed forward model parameter. Note that in $\mathbf{b}'$ only one of the parameters is perturbed at a time; for instance, when computing the systematic uncertainty related to temperature, only the temperature values are perturbed, while the rest (IWC, LWC, Rain, Snow, PSDs, etc) are left unperturbed. Then, the retrieval results using the perturbed measurements were compared to the retrieved values from an unperturbed run, i.e. where the measurements were,

$$\mathbf{y} = \mathbf{F}(\mathbf{x}_T, \mathbf{b}) \tag{10}$$

and the difference between the two was a measure of the impact of a given systematic error source. Table 2 summarizes the perturbation used.

**B  Optimal Estimation**

In this study, optimal estimation retrievals are used to estimate water vapor profiles, $\mathbf{x}$. In these retrievals the problem is ill-conditioned and to find a meaningful solution an *a*

*priori* constraint is added to the retrieval problem. Each element in the measurement vector, also denoted by $\mathbf{y}$, is determined by

$$y_{jk} = \frac{dBZ(\nu_2, r_k) - dBZ(\nu_2, z_{k-1})}{\partial r} - \frac{dBZ(\nu_1, r_k) - dBZ(\nu_1, r_{k-1})}{\partial r} \tag{11}$$

where $j$ is the frequency counter (excluding the reference frequency), $k$ is the range gate counter, and $\partial r$ is the vertical resolution. In a similar manner, the elements of the forward model are given by

$$\hat{y}_{jk} = \frac{\mathbf{F}_{\nu_2, r_k}(\mathbf{x}, \mathbf{b}) - \mathbf{F}_{\nu_2, r_{k-1}}(\mathbf{x}, \mathbf{b})}{\partial r} - \frac{\mathbf{F}_{\nu_1, r_k}(\mathbf{x}, \mathbf{b}) - \mathbf{F}_{\nu_1, r_{k-1}}(\mathbf{x}, \mathbf{b})}{\partial r}. \tag{12}$$

The solution of such system may be found by minimizing the cost function,

$$\chi^2 = [\mathbf{y} - \hat{\mathbf{y}}(\mathbf{x})]^T \mathbf{S}_y^{-1} [\mathbf{y} - \hat{\mathbf{y}}(\mathbf{x})] + [\mathbf{x}$$

$-\mathbf{a}^T \mathbf{S}_a^{-1} [\mathbf{x} - \mathbf{a}$ (13) where $\mathbf{a}$ is the *a priori* estimate with covariance $\mathbf{S}_a$. As mention before, the *a priori* used is the mean profile of 100 adjacent CloudSat ECMWF-aux water vapor values smoothed by with a boxcar average of 4 vertical level and the uncertainties in this *a priori* are assumed to be 100%. In this case, the diagonal elements of $S_y$ are given by

$$\sigma^2 = \left( \sqrt{4\delta_Z} / \partial r \right)^2 \tag{14}$$

because each element in the measurement vector involves four reflectivity measurements.

Following Rodgers (2000), the iterative solution is given by,

$$\mathbf{x}_{i+1} = \mathbf{x}_i + \left[ \mathbf{K}^T \mathbf{S}_y^{-1} \mathbf{K} + \mathbf{S}_a^{-1} \right]^{-1} \left\{ \mathbf{K}^T \mathbf{S}_y^{-1} [\mathbf{y} - \hat{\mathbf{y}}(\mathbf{x}_i)] + \mathbf{S}_a^{-1} [a - x_i] \right\} \tag{15}$$

where in this case, the Jacobian matrix is given by

$$\mathbf{K} = \frac{\partial \hat{\mathbf{y}}(\mathbf{x})}{\partial x}|_{\mathbf{x} = \mathbf{x}_i} \tag{16}$$

This technique gives an estimate of the precision in the water vapor profiles according to,

$$\mathbf{S}_x = \left(\mathbf{K}^T\mathbf{S}_y^{-1}\mathbf{K} + \mathbf{S}_a^{-1}\right)^{-1} \tag{17}$$

Another important quantity used to diagnose the retrieval performance is the "Averaging Kernel" matrix, given by

$$A = \frac{\partial \mathbf{x}}{\partial \mathbf{x}_T} = \left(\mathbf{K}^T\mathbf{S}_y^{-1}\mathbf{K} + \mathbf{S}_a^{-1}\right)^{-1}\mathbf{K}^T\mathbf{S}_y^{-1}\mathbf{K} \tag{18}$$

where $\mathbf{x}_T$ is the true state of the atmosphere and $\mathbf{x}$ is the retrieved state obtained in the last iteration of equation 15. The rows of this matrix are the averaging kernels and they map the true state into the retrieval space, that is to say, they describe how the elements of the true state influenced the retrieved state. The width of the kernel is a measure of retrieval resolution.

---

## Author Comment (AC3) · 20 May 2016

We thank Dr. Wulfmeyer for showing interest in our work. Below are our responses in blue.

**1   Dr. Wulfmeyer comment**

These results are fairly limited because obviously the authors seem to omit the analysis of noise errors in the derivation of water-vapor profiles in clouds. The DAR ist extremely sensitive to the SNR of the online and offline signals because the relative

error in the derivation of water vapor is inversely proportional to the differential optical thickness. There is a vast of literature from the investigation of water-vapor differential absorption lidar, which is nearly equivalent to DAR but is it not mentioned by the authors. I expect that it will turn out that the SNR of the return signals is far away from enabling reasonable measurements of water vapor in clouds from space. Before this analysis is done and this ommission is healed, it is hardly possible to make a reasonable judgement of this technique. I strongly recommend that the methodological analysis of DAR is extended by system noise error propagation. Therefore, also the argument that the accuracy of the measurement is increased by averaging is incorrect because the authors are dealing with systematic errors but not with uncorrelated noise. In this connection, I am wondering how useful a precision of 89% is (or an error of a factor of 2). In this case, it is probably better to guess the water vapor content of the cloud by the temperature profile. An NWP output will likely produce more accurate results.

We have the impression that Dr. Wulfmeyer believes that we omit the analysis of noise errors. In section 3, (page 4 line 18) we state that we assume a measurement noise error of 0.16dBZ, then, in section 4 and 5 we use end-to-end retrievals algorithms to propagate this error to the retrieval process and investigate its impact. This measurement noise error is similar to the one found in CloudSat (a radar from space) and it was chosen because a similar error should be achievable when such a system is implemented.

Further, Dr. Wulfmeyer is correct when he states that the accuracy will not be increased by averaging because the errors are systematic. However, we do not claim that, we clearly state that it is the precision (determined by the measurement noise) the one that will average out. For reference, in section 4 we explain the difference between precision and accuracy (systematic uncertainties), and in section 5 we state

that it is the precision that averages out.

With respect to the DIAL literature we will add the following sentence in the introduction: In this study we asses the differential absorption radar (DAR) concept to profile water vapor in cloudy and rainy areas. **This technique is analogous to the differential absorption lidar (DIAL) technique [e.g., Schotland, 1966, Browell1979, Wulfmeyer and Walther (2001)].** The DAR concept exploits the difference between the radar reflectivity at different frequencies...

**References**

Schotland, R. M.: Some observations of the vertical profile of water vapor by means of a ground based optical radar, Proceedings of the Fourth Symposium on Remote Sensing of the Environment, Environmental Research Institute of Michigan, Ann Arbor, Mich, pp. 273-283, 1966.

Browell, E. V., Wilkerson, T.D., McIlrath, T.J.: Water vapor differential absorption lidar development and evaluation, Appl Opt. 18, 20 pp,3474-3483, doi: 10.1364/AO.18.003474, 1979.

Wulfmeyer, V. and Walther, C.: Future performance of ground-based and airborne water-vapor differential absorption lidar. I. Overview and theory, Appl. Opt. 40, pp. 5304–5320, 2001.